# Radiomics-Based Deep Learning Prediction of Overall Survival in Non-Small-Cell Lung Cancer Using Contrast-Enhanced Computed Tomography

**DOI:** 10.3390/cancers14153798

**Published:** 2022-08-04

**Authors:** Kuei-Yuan Hou, Jyun-Ru Chen, Yung-Chen Wang, Ming-Huang Chiu, Sen-Ping Lin, Yuan-Heng Mo, Shih-Chieh Peng, Chia-Feng Lu

**Affiliations:** 1Department of Biomedical Imaging and Radiological Sciences, National Yang Ming Chiao Tung University, Taipei 112, Taiwan; 2Department of Radiology, Cathay General Hospital, Taipei 106, Taiwan; 3Department of Medicine, School of Medicine, Fu Jen Catholic University, Taipei 242, Taiwan; 4Department of Pulmonology, Cathay General Hospital, Taipei 106, Taiwan; 5Department of Medicine, National Tsing Hua University, Hsinchu 300, Taiwan; 6Professional Master Program in Artificial Intelligence in Medicine, College of Medicine, Taipei Medical University, Taipei 110, Taiwan

**Keywords:** radiomics, non-small-cell lung cancer, computed tomography, survival, deep learning

## Abstract

**Simple Summary:**

The five-year survival rate of non-small-cell lung cancer (NSCLC), which accounts for 85% of all lung cancer cases, is only 10–20%. A reliable prediction model of overall survival (OS) that integrates imaging and clinical data is required. Overall, 492 patients with NSCLC from two hospitals were enrolled in this study. The compensation method was applied to reduce the variation of imaging features among different hospitals. We constructed a deep learning prediction model, DeepSurv, based on computed tomography radiomics and key clinical features to generate a personalized survival curve for each patient. The results of DeepSurv showed a good performance in discriminating high and low risk of survival. Furthermore, the generated personalized survival curves could be intuitively applied for individual OS prediction in clinical practice. We concluded that the proposed prediction model could benefit physicians, patients, and caregivers in managing NSCLC and facilitate personalized medicine.

**Abstract:**

Patient outcomes of non-small-cell lung cancer (NSCLC) vary because of tumor heterogeneity and treatment strategies. This study aimed to construct a deep learning model combining both radiomic and clinical features to predict the overall survival of patients with NSCLC. To improve the reliability of the proposed model, radiomic analysis complying with the Image Biomarker Standardization Initiative and the compensation approach to integrate multicenter datasets were performed on contrast-enhanced computed tomography (CECT) images. Pretreatment CECT images and the clinical data of 492 patients with NSCLC from two hospitals were collected. The deep neural network architecture, DeepSurv, with the input of radiomic and clinical features was employed. The performance of survival prediction model was assessed using the C-index and area under the curve (AUC) 8, 12, and 24 months after diagnosis. The performance of survival prediction that combined eight radiomic features and five clinical features outperformed that solely based on radiomic or clinical features. The C-index values of the combined model achieved 0.74, 0.75, and 0.75, respectively, and AUC values of 0.76, 0.74, and 0.73, respectively, 8, 12, and 24 months after diagnosis. In conclusion, combining the traits of pretreatment CECT images, lesion characteristics, and treatment strategies could effectively predict the survival of patients with NSCLC using a deep learning model.

## 1. Introduction

In 2020, the cancer incidence and mortality data released by the American Cancer Society showed that lung cancer is the second most common malignant tumor with the highest morbidity and mortality in the United States, which accounted for 11.4% of all cancers and 18% of the total cancer deaths [1]. The 5-year survival rate of non-small-cell lung cancer (NSCLC), which accounts for 85% of all lung cancer cases, is only 10–20% [2,3]. Despite the development of state-of-the-art therapy, there has been little improvement in the survival rate. In patients with NSCLC, the same type of tumor can display different responses to therapy and prognoses in various individuals. The main factors contributing to these differences are tumor heterogeneity and the various conditions of affected patients.

Personalized medicine plays a key role in improving treatment effects; moreover, it can improve patient survival. In 2012, Lambin et al. proposed the use of radiomics to identify tumor heterogeneity [4]. Radiomic analysis is a method to mathematically extract features from medical images using statistics, imaging filters, and wavelet transforms [5]. Generally, three categories of features can be obtained, including the histogram (first-order statistics), geometry, and texture, which quantify tumor size, shape, heterogeneity, and microenvironment. In the past decade, radiomics was widely applied for diagnosis, prognosis prediction, and therapeutic response monitoring based on cancer imaging [6,7,8].

A considerable number of studies have been published on the prognosis prediction of NSCLC based on computed tomography (CT) radiomics [9]. However, some studies have demonstrated various results because of the instability and low reproducibility of radiomic features. Berenguer et al. demonstrated that CT radiomics can be affected by the tube voltage, tube current, section thickness, pixel size, reconstruction kernel, contrast enhancement, and even the use of different scanners with the same setting [10]. The lack of reproducibility and validation of radiomic studies are considered major challenges in the field. Recently, the Image Biomarker Standardization Initiative (IBSI) has established a reference framework for commonly used radiomic features to improve the robustness and application of radiomics [11]. Moreover, CT images acquired from different hospitals may have wide variations and can therefore cause potential biases in multicenter studies. Multicenter adjustment of images is essential to improve the predictive performance of applications based on radiomics. In a genomics study, to deal with batch effects caused by the acquistition from different dates, sources, and techniques, Johnson et al. introduced the ComBat compensation method to correct the variations [12]. Similar approaches have been effectively validated for positron emission tomography images [13], CT images [14,15], and even magnetic resonance images that have no standard intensity values [16]. Orlhac et al. concluded that the use of realigned features will enable multicenter studies to pool data from different sites and build reliable radiomic models based on large databases [14]. In this study, we applied our IBSI-complied radiomic analysis with ComBat compensation to the data collected from two hospitals. Accordingly, CT radiomic features can be pooled without being adversely impacted by the variability of multiple sources. To the best of our knowledge, this is the first study to present an IBSI-complied radiomics analysis combined with the ComBat compensation method to predict the survival of patients with NSCLC.

Studies on CT radiomics showed promising results in the survival prediction, treatment planning, and follow-up at all stages of NSCLC, indicating that informative imaging biomarkers could be translated into clinical practice [9]. However, most studies enrolled patients who underwent non-enhanced contrast CT (NCECT) or a mixture of both NCECT and contrast-enhanced CT (CECT) [9]. Kakino proposed that because of radiomic differences between NCECT and CECT, pooling data together may cause additional bias in NSCLC studies [17]. Gao demonstrated that CECT produces more accurate diagnoses based on the enlargement of solid portions in lung nodules and elevated Hounsfield units (HU) in ground-glass nodules [18]. In clinical practice, CECT is the main imaging modality used to diagnose NSCLC and evaluate its treatment response. However, several open-source CT databases mainly contained NCECT data, such as The Cancer Genome Atlas Lung Adenocarcinoma and The Cancer Genome Atlas Lung Squamous Cell Carcinoma. Theoretically, CECT is widely used in the routine diagnosis and regular follow-up of patients with NSCLC. Hence, we aimed to develop a CECT dataset from two local hospitals for survival prediction based on CT radiomics.

Studies have proposed the feasibility of CT radiomics to predict the overall survival (OS). Hong et al. reported that a combination of optimal radiomic signatures and clinical predictors outperformed the use of radiomics signatures only in survival prediction (concordance index [C-index] = 0.799 vs. C-index = 0.746) in NSCLC [19]. Furthermore, Yang et al. reported similar results for the combined model (C-index = 0.742) compared with the use of clinical features only (C-index = 0.617) [20]. The combination of radiomics features, a traditional staging system, and other clinicopathological risk factors could improve the predictive accuracy and treatment policies in oncology [21,22]. However, clinical data, such as the clinical stage, histological type, and tumor size, reported in previous studies are limited.

To improve the reliability of the survival prediction in NSCLC, we aimed to apply the IBSI-complied radiomics analysis on the pretreatment CECT, ComBat compensation for aligning multicenter datasets, and comprehensive clinical data as inputs for the training of a deep learning prediction model.

## 2. Materials and Methods

### 2.1. Patients

In this study, we enrolled 338 and 154 patients with NSCLC from Cathay General Hospital (CGH) and Sijhih Cathay General Hospital (SCGH), respectively. The inclusion criteria were as follows: (a) patients with histologically or cryptologically confirmed NSCLC; (b) those who underwent CECT before treatment; (c) those whose image quality was sufficient to identify and quantify the tumors; and (d) those for whom survival time was recorded. Pretreatment CECT and clinical data, including age, gender, clinical stage, OS time, survival status, and treatment, were collected. This study was approved by the Institutional Review Board of CGH (CGHIRB: CGH-P109087), and the requirement of informed consent was waived because of the retrospective nature of this study.

### 2.2. Computed Tomography Data Acquisition

Patients from CGH underwent CECT protocols using a 64-slice multidetector CT (MDCT) scanner (Brilliance 64, Philips Healthcare, Amsterdam, The Netherlands) or 320-slice dynamic volume CT (Aquilion ONE; Toshiba Medical Systems, Otawara-shi, Japan). CECT images collected from SCGH were acquired using a 16-slice MDCT scanner (Somatom Emotion 16, Siemens Healthcare, Erlangen, Germany). For both hospitals, the CT scan coverage was similar, ranging from the thoracic inlet to the upper abdomen. CT images were reconstructed with a slice thickness of 5 mm. Pixel sizes ranged from 0.35 × 0.35 mm^2^ to 0.58 × 0.58 mm^2^. Each slice had a matrix size of 512 × 512 pixels with a 16-bit gray-scale resolution in HU. The peak tube voltage was 120 kVp, and the tube current was decided by automatic dose modulation ranging from 60 to 200 mA.

### 2.3. Tumor Segmentation and Radiomic Feature Extraction

The workflow of the building prediction model is displayed in Figure 1. For patients with multiple lesions, the largest tumor was selected as the target lesion. For each patient, the regions of interest (ROIs) were delineated by a well-trained radiological technologist and checked by one of three experienced radiologists. The image resolution was adjusted to resample all voxel sizes as an isotropic 2 × 2 × 2 mm^3^. Wavelet analysis with a three-dimensional low (L) or high (H) spatial frequency filter was applied on the preprocessed images. The radiomic features, including 16 first-order (histogram) features, eight shape-and-size features, and 49 textural features (including a gray-level co-occurrence matrix, gray-level run-length matrix, and local binary pattern [LBP]) with a discretization of 32 bins, were calculated on the original CECT and eight wavelet-filtered images. Overall, 593 features were acquired to quantify tumor characteristics for each patient with NSCLC. The image preprocessing and extraction of radiomic features were performed using our previously published multimodal radiomics platform (available online: https://cflu.lab.nycu.edu.tw/MRP_MLinglioma.html (accessed on 1 June 2022), which complied with the IBSI on the MATLAB R2020a environment [7].

### 2.4. Realignment of Multicenter Radiomic Datasets

The multicenter effect was evaluated by comparing the distribution of each CECT radiomic feature between the two hospitals. The feature distribution, y, can be represented as follows (1): y_ijk_ = α_k_ + X_ij_β_k_ + γ_ik_ + δ_ik_ε_ijk._(1)
where i indicates the ith hospital, j indicates the jth patient, and k represents the kth features; α is the average of each feature; X indicates the biological covariates; β is the linear regression coefficient for the covariant matrix X; γ estimates the multicenter effect; δ measures the effects of various scanners and protocols; and ε indicates individual variance. To reduce (compensate for) the multicenter effect, the ComBat compensation method was applied [13,14,15,16]. The following equation was used to realign feature distribution (2):
(2)$$y_{\mathrm{ijk}}^{\mathrm{ComBat}}=(y_{\mathrm{ijk}}-\alpha e^{k}-X_{\mathrm{ij}}\beta e^{k}-\gamma e^{\mathrm{ik}})/\delta e^{\mathrm{ik}}+\alpha e^{k}$$

Estimators, αe^k^, βe^k^, γe^ik^, and δe^ik^, were calculated based on the maximum likelihood. To correct the multicenter effect, the realigned features ($y_{\mathrm{ijk}}^{\mathrm{ComBat}}$) were calculated by adding the personal variance (ε_ijk_) to the average value (αe^k^). In this study, we considered the clinical stage as the biological covariant for the realignment. The processing was performed using the ComBat function developed by Orlhac et al. on MATLAB R2020a [16].

### 2.5. Selection of Survival Predictors

After the realignment of the CGH and SCGH datasets, 30% of the patients were assigned as the test dataset. The remaining data were used as the training dataset for the subsequent feature selection and model training. This study was designed to predict the survival probability of patients with NSCLC 8, 12, and 24 months after diagnosis. To effectively differentiate the survival status during the 8 and 24 months, we performed the feature selection based on the patients with OS within this period (n = 169) in the training dataset. In the feature selection, the univariate Cox regression ($p\leq 0.1,\;\left|b\right|\geq 2$) was used to select radiomic features for survival prediction. The coefficient, b, was used to estimate the Cox proportional hazard of OS duration on each radiomic feature. A large absolute value of b indicates a high influence on survival. Patients with an OS longer than 12 months were defined as the good survival group; those with an OS shorter than 12 months were assigned to the poor survival group. The categorical clinical features were selected based on the chi-square test (p≤0.1) regarding the survival groups. Then, we input the selected features to train the prediction model using the full training dataset (n = 345).

### 2.6. Survival Prediction with Deep Neural Networks

A prediction model was constructed based on the DeepSurv architecture, a deep neural network, using R programming (version 4.1.1). DeepSurv integrated the Cox proportional regression and deep neural network, which contained input, multilayer hidden, and output layers [23]. The prediction model output the hazard risk function to generate personalized survival curves. The coefficients of the hazard risk function for predictors were determined based on the weights of hidden layers. The average negative log partial likelihood of the Cox function was applied as the loss function to optimize the network (3): (3)$$l(\theta )=\sum_{\left(i:E=1\right)}(he^{\theta }(X_{i})-\log(\sum_{j\epsilon R(\mathrm{Ti})}\cdot e^{\left(he^{\theta }(\mathrm{xj})\right)})/N_{E}=1+\lambda \times \theta_{2}^{2}$$
where x is the input features, N_E=1_ indicates the number of patients with an observable event, R(T_i_) presents patients with survival longer than T_i_, λ is the L2 regularization parameter, and θ represents the model weights. Gradient descent optimization was used to minimize l(θ).

We developed a DeepSurv model with a five-layer neural network to predict the hazard function. Fully connected layers with 20, 26, 32, 26, and 20 nodes were applied in five hidden layers, respectively. We applied a rectified linear unit as the activation function, followed by a batch normalization layer to avoid the risk of internal covariate shift for each hidden layer. The dropout rate for the input and hidden layers was 0.4 to prevent overfitting. The following parameters were applied for model training: initial learning rate, 0.01; epoch number, 400; momentum, 0.8; and the Adam optimizer.

### 2.7. Statistical Analysis

To assess the inter-observer variation of ROI delineation, 20 NSCLC tumors were randomly selected, and the ROIs of each tumor were separately delineated by the three radiologists. The intraclass correlation coefficient (ICC) and overlapping rate were subsequently calculated (Appendix A), and the ICC of radiomic features were also provided (Appendix A). The ICC was applied to measure the observation bias in radiomic features. Radiomic features with a high ICC (R≥0.75) indicated high consistency among the three observers. The overlapping rate indicated the observation bias in contours among the three observers.

To assess the personalized survival curve to differentiate the high and low risk of OS, the test dataset was divided into two groups based on the 12-month OS. We calculated the mean and standard deviation of the predictive survival probability of each group, and the log-rank test was used to investigate the difference in personalized survival curves between the two groups. To evaluate the overall performance of the prediction model, the C-index and time-dependent receiver operating characteristic (ROC) curves were estimated at 8, 12, and 24 months. The optimal thresholds could be further determined based on the time-dependent ROC curves for three selected time points, respectively. Then, the reference risk curve was constructed based on the curve fitting passing through the three time-dependent thresholds using the Weibull probability distribution function [24]. The risk levels of death could be estimated using the area between the personalized survival curve and reference risk curve at the following periods: <8, 8–12, 12–24, and >24 months. A negative risk level represented that the segment of the personalized survival curve was below the reference risk curve, indicating a high risk of event occurrence within the period. The prediction performance among the combined (radiomic + clinical), radiomic, and clinical models was assessed and compared using AUC and C-index values with the bootstrap resampling 100 times, respectively.

## 3. Results

### 3.1. Demographic Data of Multicenter Datasets

The clinical data collected from the CGH and SCGH are listed in Table 1. The data, such as the tumor laterality, clinical T stage, clinical stage, histology, surgery, and survival status, showed no significant difference between the two hospitals. However, data on radiotherapy, alcohol use, betel nut use, and the survival period showed significant differences between the two hospitals. The CGH dataset included patients with better survival (*p* < 0.001) compared with the SCGH dataset. The identified significant differences between the two hospitals reflected data variation, which could be commonly observed in multicenter studies and real-life situations. This phenomenon highlighted the importance of building a prediction model by pooling multicenter datasets with sufficient variation to ensure model generalization. The complete demographic data from CGH and SCGH are listed in Appendix A.

### 3.2. Variation Estimation of Lesion Contouring and Radiomic Features

The overlapping rate was used to determine the observation bias of lesion contouring. Almost all patients showed high consistency in lesion contouring (overlapping rate = 0.78 ± 0.18). We demonstrated the image and contour of the 20 patients in Appendix A. The observation bias in radiomic features was evaluated using the ICC. We calculated the mean and standard deviation of the ICC of 19 feature categories (Appendix A). All radiomic features showed sufficiently high consistency with average ICC values of greater than 0.819 among the three observers. Texture features with LLL, LLH, LHL, HLH, and HHH wavelet filters also exhibited high consistency with ICC values of greater than 0.868 among the three observers. These results indicated that the radiomic features are highly stable.

### 3.3. Selected Radiomic and Clinical Features for the Prediction Model (ICC)

Eight radiomic and five clinical features were selected as predictors for model building (Table 2). Radiomic predictors were selected from texture features with LLL, LLH, LHL, HLL, HLH, and HHH wavelet filters. Positive b values were observed in LHL_Homogeneity (*b* = 2.0202), HLL_Homogeneity (*b* = 2.5763), HLH_Inverse variance (*b* = 3.4462), and HHH_Correlation (*b* = 3.0696), which indicated negative correlations between these features and OS. Radiomic features, including LLL_LBP_Uniformity and Short Run Emphasis with LLH, HLL, and HLH wavelet filters, showed negative b values of less than −2.2332. Negative b values indicated decreased hazard and increased survival times. Clinical features, including histology, clinical T stage, clinical N stage, clinical stage, and surgery, were selected in model building.

### 3.4. Representative Cases for Predicting Personalized Survival Curves

Figure 2 shows the CT images, predicted survival curves, and risk levels of five representative cases. The reference risk curve (dashed curve) was also plotted to intuitively represent the risk of death. Case #1, with an OS of 3 months, had a 3.5-cm lesion located in the left upper lobe with adjacent pleural thickening. The predicted survival curve generated by the DeepSurv model was far below the reference risk curve, indicating poor survival. The risk level of Case #1 showed a negative value (−0.54) in the first period (<8 months) (Figure 2f). Case #3, with an OS of 17 months, had a 3-cm mass located in the left upper lobe with a shadow with small air density. The predicted survival curve generated by the DeepSurv model intersected the reference risk curve at the 12–24-month period, demonstrating poor survival over the duration (Figure 2c). The risk level for Case #3 showed a negative value (−0.60) at the third period (12–24 months) (Figure 2f). Case #5, with an OS of 68 months, had a solitary lobulated pulmonary nodule located in the left upper lobe. The predicted survival curve generated by the DeepSurv model was far above the reference risk curve, revealing good survival (Figure 2e). The risk level for Case #5 showed positive values at the four periods (Figure 2f).

### 3.5. Superior Performance of the Prediction Model Based on Combined Features

To compare model performance based on combined (radiomic + clinical), radiomic, and clinical feature sets, we applied a bootstrap resampling on the test dataset 100 times. The combined model (C-index = 0.74, 0.75, and 0.75 at 8, 12, and 24 months, respectively) significantly outperformed the clinical (C-index = 0.70, 0.70, and 0.71, respectively) and radiomic (C-index = 0.64, 0.65, and 0.65, respectively) models, indicating that combining radiomic predictors with clinical predictors could improve survival prediction (Figure 3a). Similarly, the combined model displayed significantly higher AUCs at the three time points (AUCs = 0.76, 0.74, and 0.73, respectively) than the radiomic (AUCs = 0.66, 0.66, and 0.63, respectively) and clinical (AUCs = 0.69, 0.64, and 0.67, respectively) models, reflecting an improvement in the prediction efficacy (Figure 3b).

The detailed performance of the combined model is displayed in Figure 4. Figure 4a shows the time-dependent ROC curves, AUC values, sensitivity, specificity, and optimal probability threshold at the three time points. The predicted survival curves for the poor survival group (blue line and area) quickly dropped down at the beginning (Figure 4b). In contrast, the predicted survival curves for the good survival group (red line and area) showed a gradual reduction in survival probability, indicating a better prognosis than the poor survival group. A significant difference in the mean predicted survival curves was observed between the two groups (*p* = 0.0015).

## 4. Discussion

This study included 492 patients with NSCLC at all stages. In our results, the performance (estimated by the C-index and AUC) of the model combining clinical and radiomic features is better than that of the models using radiomic or clinical features alone at the three time points. We analyzed similar studies that included all NSCLC stages; Huang et al. developed models that predicted OS in the anaplastic lymphoma kinase (ALK+) set, which achieved a C-index of 0.649 (95% confidence interval, 0.640–0.658) [25]. Timmeren et al. proposed a radiomic model that contained only radiomic features; this model reached a C-index of 0.63 for OS prediction using cone beam CT [26]. Yang et al. compared the performance of 2D and 3D radiomics in predicting the survival of patients with NSCLC, and the results demonstrated that the combined 2D and 3D features (C-index = 0.742) showed a better prognostic performance than 2D (C-index = 0.687) or 3D features alone (C-index = 0.736) [20]. Li et al. applied a predictive nomogram in the external test cohort, and the C-index reached 0.660 [27]. In this study, we included more patients with comprehensive clinical and radiomic features to predict personalized survival curves with a superior performance.

Survival prediction in NSCLC is important but challenging for patients and physicians. In this study, we proposed a deep learning prediction model to evaluate the survival probability. The superior performance of our proposed model may be attributed to the following factors. First, the features extracted from CECT were more discriminate in outcome prediction than those extracted from NCECT [28]. Second, the comprehensive clinical features (i.e., age, gender, clinical stages, histological types, tumor size, smoking status, and alcohol use) combined with quantitative radiomics included in this study can build a dependable prediction model. Third, our image preprocessing and feature extraction methods were in accordance with the IBSI recommendations [16]. Because of the standardization of radiomics, the reliability and reproducibility of the prediction models showed a satisfactory performance. Fourth, the multicenter dataset for assessing model performance ensured the generalization of the results of this study for future applications. Besides, the ComBat compensation method further compensated for the variations in datasets acquired from different sources. Finally, the applied DeepSurv model incorporated the multilayer architecture to handle high-dimensional feature sets, resulting in a superior performance in survival prediction.

Radiomics has been used to explore tumor heterogeneity, pattern, and microenvironment and is promising in assessing and predicting histopathological characteristics, treatment response, and clinical outcomes in NSCLC [9,11,20]. The performance of radiomics analysis based on CECT images could provide the density distribution on the intratumoral physiology of blood supply. Chen et al. proposed that CECT images showed better diagnostic capability than NCECT images because CECT images can supply more information on intratumoral microvascular density [29]. This study also demonstrated that CECT radiomics combined with clinical features could discriminate prognosis in NSCLC. Additionally, most hospitals considered CECT as the primary protocol for regular follow up in NSCLC.

In this study, extraction of radiomic features only from the largest lesion was attributed to three reasons. First, multiple lesions might possess different image traits. Using an appropriate aggregation method of radiomic features extracted from different lesions is critical for subsequent model building. A recent study proposed by Chang et al. investigated four aggregation methods of radiomics, including the largest lesion, weighted average of three largest lesions based on volume, weighted average of all lesions, and simple average of all lesions, for patients with brain metastases [30]. They concluded that the largest lesion and the weighted average of the three largest lesions provided the best performance of survival prediction. This result indicated that radiomic features extracted from the largest lesion might contain sufficient and critical information for survival prediction. Considering only the largest lesion (usually the most representative one) could further eliminate the potential variation and “averaging effect” of the feature aggregation from multiple lesions. Second, the clinical data such as the histological type and clinical T stage were mostly investigated based on the largest/primary tumor lesion. Accordingly, combining the radiomic features extracted from the largest lesion and clinical data provided good consistency to characterize the patient status. Finally, our combined model also included the clinical T stage and overall clinical stage. These two clinical features reflected the presence of multiple lesions and invasion of the disease. Based on the abovementioned reasons and our results, we suggested that the radiomic features based on the largest lesion and clinical staging data could provide critical information for survival prediction.

Survival prediction models combining radiomic and clinical features outperformed the model only depending on clinical or radiomic features alone. Regarding clinical features, five clinical features, including histology, clinical T stage, clinical N stage, clinical stage, and surgery (Table 2), were selected as survival predictors. However, the use of chemotherapy was not selected. This might be because the chemotherapy guidelines have varied during the past decade due to the rapid development of chemotherapy regimens. The development of systemic therapy grew quickly in chemotherapy, targeted therapy, and immunotherapy. Nowadays, the guidelines for chemotherapy are still undergoing dynamic changes in NSCLC [31]. Due to the different implementation in chemotherapy regimens, we could only stratify the application of chemotherapy into “yes” or “none” in our study. This binary stratification might cause the result that chemotherapy showed no significant difference (*p* = 0.20) between the good and poor survival groups and therefore was not selected as a predictor. Regarding radiomic features, eight texture features were selected to build prediction models. Studies had proposed that texture features are correlated with OS because they can reflect tumor heterogeneity [28,32,33]. These findings are consistent with those reported in this study. With the good reflection of tumor heterogeneity, texture features have higher survival predictive power in NSCLS [34], particularly those with wavelet filters.

Based on previous studies and experience in clinical practice, multiple factors could interactively influence patient survival. We took the targeted therapy as the example. Appendix A compares the clinical features between patients with and without targeted therapy. In the targeted therapy group, 89.6% of patients (155 of 173 patients) belonged to the adenocarcinoma type, 92.5% of patients (160 of 173 patients) had stage IV NSCLC, 93.1% of patients (161 of 173 patients) did not receive surgery, and 70.5% of patients (122 of 173 patients) had a known epidermal growth factor receptor (EGFR) mutation. These ratios were all significantly higher (*p* < 0.001) than those in the non-targeted therapy group. Even though the targeted therapy might benefit the outcome in the advanced NSCLC group with an EGFR mutation, the higher clinical stage and inaccessible to surgery limited patient survival. Accordingly, no significant difference (*p* = 0.13) in OS between targeted therapy (19.51 ± 19.01 months) and non-targeted therapy (16.93 ± 17.55 months) groups was observed. Instead of focusing on a single factor, we collected comprehensive data to evaluate the overall/interactive effects of features on OS. We believed that the proposed prediction model based on the clinical and imaging features for patients with stage I to IV NSCLC could benefit the clinical management.

The robustness of reliability and reproducibility is the main problem in implementing diagnosis or treatment prediction in clinical practice [35]. Our systems of image preprocessing and feature extraction were in accordance with the IBSI recommendations [11]. The IBSI provided a standard reference for each essential step in radiomics analysis, including image preprocessing, lesion segmentation, feature extraction, and validation. In this study, the ICC of each radiomic feature among the three observers was higher than 0.75, and the ROI overlapping rate was high (78%), indicating the robustness of radiomics. Recently, the Food and Drug Administration has declared 10 guiding principles for evaluating good machine learning practice [1]. One of the principles is that training datasets are independent of test datasets, indicating that training and test datasets are selected and maintained to be appropriately independent of one another [36]. Besides, evaluating the performance of artificial intelligence devices across multiple clinical sites is important to ensure that the algorithms perform well across reprehensive populations [1]. Another technical consideration of multicenter studies is the data heterogeneity between different sites. With the ComBat compensation method, datasets from two sites/scanners can be adjusted to reduce potential biases. To our knowledge, our method is the first to combine the IBSI and ComBat compensation method to standardize radiomics. In this study, the performance of the training and test datasets were good, indicating that our algorithm is stable.

DeepSurv is a modern Cox proportional hazards, deep neural network proposed by Katzman in 2018 [23]. DeepSurv can predict model interactions between a patient’s covariates and treatment effectiveness, which can provide personalized treatment recommendations. Their results showed that DeepSurv outperformed other survival analysis methods on survival data with linear or nonlinear effects from covariates [23]. In our results, DeepSurv showed good performance in discriminating high and low risk of survival. The reason for this is that DeepSurv could combine clinical and radiomic features to develop a superior model. Personalized survival curves could be applied for individual survival prediction in clinical practice. Clinical physicians could make therapeutic plans in reference to predictive survival curves before treatment begins. For patients whose predictive survival curve was under the reference risk curve in the early period, indicating a poor prognosis (Figure 2a), modification of their treatment strategy should be considered by their physicians. Otherwise, if patients’ predictive survival curve was above the referenced risk curve, physicians would be more confident of a good prognosis from the arranged treatment (Figure 2e). In summary, personalized survival curves derived from a DeepSurv model combining both radiomic and clinical features may improve the clinical management of patients with NSCLC.

This study had several limitations. First, our samples included uneven patient populations in terms of clinical stage of NSCLC. High clinical stages of IV and III comprised 70–80% of the patients in our study cohort. This was because most patients underwent CT examination and received subsequent treatment after the occurrence of symptoms. Second, our prediction model did not include the patients’ genetic profile, such as the EGFR and anaplastic lymphoma kinase (ALK) status. Only 52% and 14% of the recruited patients had the EGFR mutant and ALK status, respectively. The main reason for the shortage of this information was because the percentage of adenocarcinoma in these two hospitals was only approximately 60–70%. The EGFR gene test was mainly performed in patients with adenocarcinoma. Third, our validation had not included international datasets. Considering that most open-source databases included only NCECT datasets, future data collection of CECT may be required to further validate our proposed model. Another challenge is that our study included complete clinical data; however, most open-source databases included limited clinical information. Future studies with a comprehensive clinical dataset and international CECT database should be considered to generate a more generalized model for clinical implementation.

## 5. Conclusions

In conclusion, this study employed robust and reproductive radiomics combined with clinical data to build a DeepSurv prediction model. We found that personalized medicine based on the OS prediction using DeepSurv was feasible, which could benefit physicians, patients, and caregivers in managing NSCLC.

## Figures and Tables

**Figure 1 cancers-14-03798-f001:**
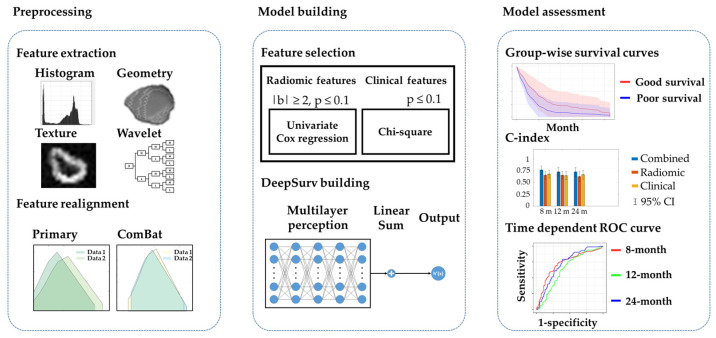
Workflow of the prediction model. First, radiomic features were extracted from CT images using multimodal radiomics platform. To remove the centric effect, we applied ComBat realignment on the radiomic features. Second, the dimension feature was reduced with feature selection to avoid the overfitting effect. The remained features after feature selection were applied to build a model based on DeepSurv. Finally, the assessment of model performance relied on group-wise survival curves, C-index, and time-dependent ROC curves.

**Figure 2 cancers-14-03798-f002:**
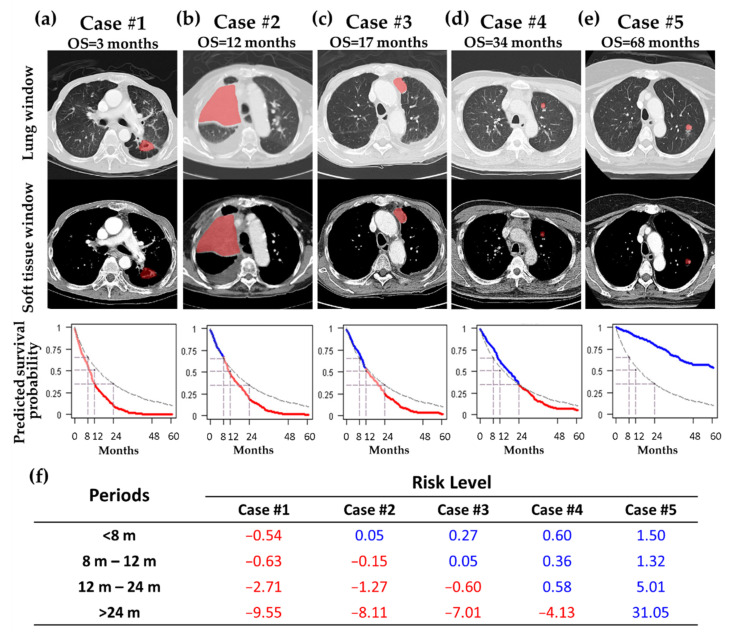
Demonstration of the predictive survival curve, AUC difference, and CT images in five patients. Five predictive survival curves (**a**–**e**) were displayed with OS of 3, 12, 17, 34, and 68 months. The first and second rows display lung and soft tissue windows of lung CT images (pink ROIs indicate the segmented tumors). The third row shows personalized survival prediction (color line) and referenced risk curves (dash line). For the colored line, blue segments indicate that the patients were at a low risk during the period, and red segments demonstrate that the patients were at a high risk during the period. (**f**) lists the risk levels of the five patients during four periods (<8, 8–12, 12–24, and >24 months).

**Figure 3 cancers-14-03798-f003:**
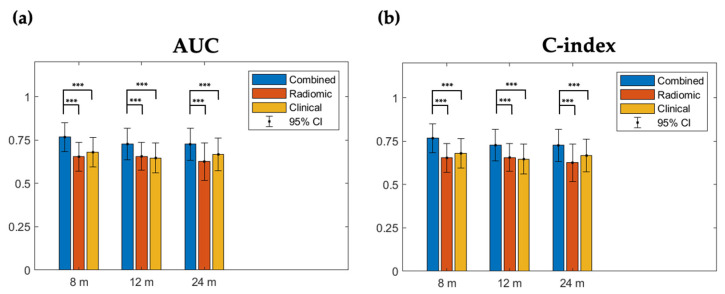
Comparison of model performance among the combined, radiomic, and clinical models. AUCs and C-indices based on the three optimal probabilities were evaluated with bootstrap resampling 100 times. Red, green, and blue lines correspond to C-index from the combined, radiomic, and clinical models, respectively. (**a**) AUCs in the combined model have the best performance at three time points. The combined model showed significantly higher AUCs than the other models. (**b**) C-indices in the combined model have the best performance at three time points. The combined model showed a significantly higher C-index than the other models. 95% CI: 95% confidence interval; Red: 8 months; Green: 12 months; Blue: 24 months; ***: *p* < 0.001.

**Figure 4 cancers-14-03798-f004:**
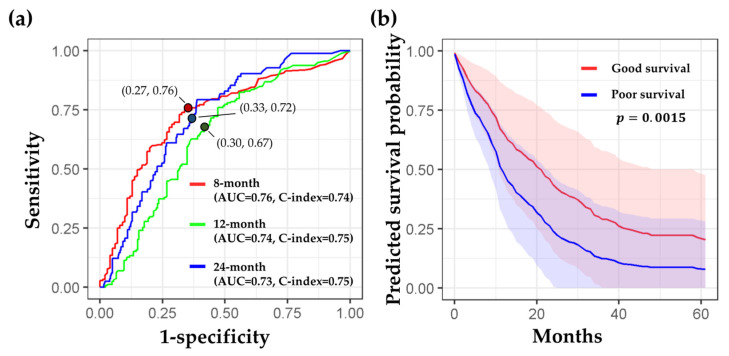
Detailed performance of the combined model (radiomic + clinical features). (**a**) Time-dependent ROC curves at 8, 12, and 24 months. Red, green, and blue lines correspond to ROC curves while predicting patients’ survival duration longer than 8, 12, and 24 months, respectively. The red, green, and blue dots are located with the values of “1-specificity” and “sensitivity,” which were estimated with the three optimal probability thresholds (0.65, 0.51, and 0.35) at 8, 12, and 24 months, respectively. (**b**) The mean survival curves from the good (red) and poor (blue) survival groups were displayed. The significance of the predicted survival probability between the two groups was assessed using a log-rank test.

**Table 1 cancers-14-03798-t001:** Characteristics of non-small-cell lung cancer cases from CGH and SCGH.

Characteristics	CGH (n = 338)	SCGH (n = 154)	*p* Value
**Age (y)**	66.98 ± 12.24	66.62 ± 11.84	0.56
**Gender**			0.75
Male	183 (54.14%)	81 (52.60%)	
Female	155 (45.86%)	73 (47.40%)	
**Histology**			0.16
Adenocarcinoma	218 (64.50%)	111 (72.08%)	
Squamous cell carcinoma	73 (21.60%)	23 (14.94%)	
Adenosquamous carcinoma	11 (3.25%)	1 (0.65%)	
Large cell cancer	4 (1.18%)	2 (1.30%)	
Other NSCLC	32 (9.47%)	17 (11.04%)	
**Clinical T stage**			0.47
0	1 (0.30%)	0 (0.00%)	
1	40 (11.83%)	18 (11.69%)	
2	93 (27.51%)	32 (20.78%)	
3	70 (20.71%)	39 (25.32%)	
4	134 (39.64%)	65 (42.21%)	
**Clinical N stage**			0.59
0	108 (31.95%)	54 (35.06%)	
1	27 (7.99%)	11 (7.14%)	
2	85 (24.85%)	44 (28.57%)	
3	118 (34.91%)	45 (29.22%)	
**Clinical M stage**			0.07
0	110 (32.54%)	63 (40.91%)	
1	228 (67.46%)	91 (59.09%)	
**Clinical stage**			0.16
I	48 (14.20%)	23 (14.94%)	
II	12 (3.55%)	11 (7.14%)	
III	50 (14.79%)	29 (18.83%)	
IV	228 (67.46%)	91 (59.09%)	
**Surgery**			0.66
None	250 (73.96%)	111 (72.08%)	
Yes	88 (26.04%)	43 (27.92%)	
**Chemotherapy**			0.06
None	202 (59.76%)	78 (50.65%)	
Yes	136 (40.24%)	76 (49.35%)	
**Radiation therapy**			0.02 *
None	217 (64.20%)	115 (74.68%)	
Yes	121 (35.80%)	39 (25.32%)	
**Targeted therapy**			0.36
None	215 (63.61%)	104 (67.53%)	
Yes	123 (36.39%)	50 (32.47%)	
**Smoke**			0.91
None	188 (55.62%)	88 (57.14%)	
Yes	138 (40.83%)	66 (42.86%)	
Not available	12 (3.55%)	0 (0.00%)	
**Betel nut use**			<0.001
None	300 (88.76%)	114 (74%)	
Yes	27 (7.99%)	22 (14.3%)	
Not available	11 (3.25%)	18 (11.7%)	
**Alcohol use**			<0.001
None	240 (71.01%)	92 (59.74%)	
Quit drinking	33 (9.76%)	18 (11.69%)	
Sometimes	20 (5.92%)	11 (7.14%)	
Always	34 (10.06%)	15 (9.74%)	
Not available	11 (3.25%)	18 (11.69%)	
**Survival Status**			0.20
Alive	97 (28.70%)	53 (34.42%)	
Dead	241 (71.30%)	101 (65.58%)	
**Duration (months)**	20.15 ± 19.95	12.75 ± 11.70	<0.001
Median	12	11	

Note: *p* values are the results of the χ^2^ test for categorized variables and t-test for continuous variables. * *p* < 0.05.

**Table 2 cancers-14-03798-t002:** Predictors for model building. Eight radiomic features and five clinical features were selected as predictors for model building.

Radiomic Features	Good Outcome (n = 83)	Poor Outcome (n = 86)	*b* Value	*p* Value
LLL_LBP_Uniformity	0.18 ± 0.05	0.20 ± 0.05	−4.57	0.03
LLH_Short Run Emphasis	0.86 ± 0.08	0.88 ± 0.08	−2.23	0.04
LHL_Homogeneity 1	0.46 ± 0.11	0.40 ± 0.12	2.02	0.02
HLL_Homogeneity 1	0.41 ± 0.09	0.37 ± 0.09	2.58	0.01
HLL_Short Run Emphasis	0.90 ± 0.05	0.92 ± 0.04	−4.22	0.04
HLH_Inverse variance	0.34 ± 0.07	0.29 ± 0.09	3.45	0.01
HLH_Short Run Emphasis	0.90 ± 0.04	0.92 ± 0.04	−4.77	0.02
HHH_Correlation	0.04 ± 0.06	0.05 ± 0.06	3.07	0.05
**Clinical Features**	**Good Outcome** **(n = 83)**	**Poor Outcome** **(n = 86)**	**chi^2^ Value**	** *p* ** **Value**
Histology	1 [1–2]	1 [1–2]	8.22	0.08
Clinical T stage	3 [2–4]	3 [2–4]	50.47	<0.001
Clinical N stage	2 [2–3]	1 [0–2]	15.74	0.03
Clinical stage	IV [III–IV]	IV [II–IV]	17.17	0.02
Surgery	0 [0–0]	0 [0–1]	7.18	0.07

Cox regression was applied on radiomic features, and clinical features were assessed using the chi-square test. The b coefficients were assessed by Cox regression. Negative coefficients indicated decreased hazard and increased survival times. L represents a low-pass filter, and H represents a high-pass filter of the wavelet decomposition. The combination of L and H letters stands for the filter type applied to the three image axes in order. LBP, local binary pattern.

## Data Availability

All data generated or analyzed during this study are included in this published article and its Supplementary Material Files.

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
