# Peer review of "Radiomics-Based Deep Learning Prediction of Overall Survival in Non-Small-Cell Lung Cancer Using Contrast-Enhanced Computed Tomography"

_cancers, 2022, doi:10.3390/cancers14153798_

Round 1

Reviewer 1 Report

Very interesting and well-written paper. Survival prediction in NSCLC patinets is very hard in a clinical setting. Prediction models have been used in the past, however the authors show outstanding results with their combined technology. The results show that the combined use of clinical features and radiomics for prediction of survival had superior results than the models alone.  

Abstract: 

One misspelling error: 

line 21:  „deep leaning” instead of deep learning

The introduction is well-written, easy to follow and contains all relevant information and references. 

The methods and results sections are also clear and well-written. I have the following questions:

Line 143: it is stated that in case of multiple lesions the largest was selected as target. My question is: how can the software predict survival based only on the largest lesion? Multiple lesion cancers are in higher stage with known worse survival. 

Line 192: why did the authors use p<0.1 instead of the more widely used p<0.05?

Results:

Line 276-278: How come the use of chemotherapy was not selected as a factor for model building?

Reviewer 2 Report

1. Why is b value given in Table 2 for radiological characteristics and chi2 value for clinical characteristics? 2. Multivariate analysis of both radiological and clinical characteristics was not performed. 3. In general, it is not clear how the selection of features was carried out, why the histological type was not included, gender/age, etc. Give the results of the selection. 4. It seems to me that radiological signs should correlate with the stage of the disease, i.e. are not independent. Was there a check for the presence/absence of correlations? 5. In general, the work seems rather superficial, for example, in patients with adenocarcinoma with an EGFR mutation during targeted therapy, the prognosis may be completely different. How can this be taken into account in your model?

Round 2

Reviewer 2 Report

The authors partially answered the questions of the reviewer. However, given the marginal gain in prediction accuracy when both radiological and clinical data are used, wouldn't it make sense to add some other data, such as blood chemistry, etc.? I still believe that the chosen clinical parameters are not independent (T stage for example is related to the clinical stage, etc.), and they are also associated with radiological parameters, which the authors confirmed in Figure 2.4. Is there any data redundancy in this case? and what is the real benefit of the resulting prognosis for clinical practice, if important factors of treatment and genetic heterogeneity are not taken into account?

Author Response

We thank you for your comments. Please see the attachment.

Round 3

Reviewer 2 Report

I have no more comments on the article. I believe that in its present form the article can be recommended for publication.